# Thoracic Radiotherapy in Extensive Disease Small Cell Lung Cancer: Multicenter Prospective Observational TRENDS Study

**DOI:** 10.3390/cancers15020434

**Published:** 2023-01-10

**Authors:** Salvatore Cozzi, Alessio Bruni, Maria Paola Ruggieri, Paolo Borghetti, Vieri Scotti, Davide Franceschini, Michele Fiore, Maria Taraborrelli, Fabrizio Salvi, Marco Galaverni, Luisa Savoldi, Luca Braglia, Andrea Botti, Sebastiano Finocchi Ghersi, Giaj-Levra Niccolò, Frank Lohr, Cinzia Iotti, Patrizia Ciammella

**Affiliations:** 1Radiation Oncology Unit, Azienda USL-IRCCS di Reggio Emilia, 42123 Reggio Emilia, Italy; 2Radiation Oncology Department, Centre Lèon Bèrard, 693736 Lyon, France; 3Radiation Therapy Unit, Department of Oncology and Hematology, University Hospital of Modena, 41125 Modena, Italy; 4Radiation Oncology Department, ASST Spedali Civili and University of Brescia, 25123 Brescia, Italy; 5Radiation Oncology Unit, Oncology Department AOU Careggi Firenze, 50134 Florence, Italy; 6Radiation Therapy Unit, IRCCS Humanitas Cancer Center, 20089 Milan, Italy; 7Radiation Oncology, Campus Bio-Medico University, Fondazione Policlinico Universitario Campus Bio-Medico, 00128 Rome, Italy; 8Radiation Oncology Unit, “SS Annunziata” Hospital, “G. D’Annunzio” University, 66100 Chieti, Italy; 9Radiation Oncology Unit, Bellaria Hospital, 40139 Bologna, Italy; 10Radiotherapy Unit, Azienda Ospedaliera Universitaria, 43126 Parma, Italy; 11Research and Statistics Infrastructure, Azienda Unità Sanitaria Locale–IRCCS di Reggio Emilia, 42123 Reggio Emilia, Italy; 12Medical Physics Unit, Azienda USL-IRCCS di Reggio Emilia, 42123 Reggio Emilia, Italy; 13Radiation Oncolgy Unit, AOU Sant’Andrea, Facoltà di Medicina e Psicologia, Università La Sapienza, 00185 Rome, Italy; 14Advanced Radiation Oncology Department, IRCCS Sacro Cuore Don Calabria Hospital, Cancer Care Center, 37024 Verona, Italy; 15Department of Medical and Surgical Science, University of Modena and Reggio Emilia, 41125 Modena, Italy

**Keywords:** consolidative radiotherapy, SCLC, small cell lung cancer, immunotherapy, radiotherapy, target therapy, lung cancer

## Abstract

**Simple Summary:**

Small cell lung cancer (SCLC) accounts for about 15% of all lung cancers, and seventy percent of patients already have advanced disease at diagnosis. In advanced disease, the use of consolidative chest RT should be recommended for patients with good response to platinum-based first-line chemotherapy, but its use has not yet been standardized. This prospective study was carried out with the intention of evaluating the spread in Italy of the use of thoracic RT in ES-SCLC, with a focus on the pattern of care (RT modalities, volumes and doses) and its effectiveness in terms of disease control and tolerability. From January 2017 to December 2019, sixty-four patients were enrolled. An extensive variability in doses, treatment volume and technique were recorded. Nevertheless, consolidative RT was well-tolerated by all patients and, after treatment, over 66% of patients did not experience in-field progression, and it has been shown to be useful in reducing the risk of thoracic disease progression in patients with advanced stage SCLC, with good response after first-line chemotherapy.

**Abstract:**

(1) Introduction: Small cell lung cancer (SCLC) is an aggressive tumor type, accounting for about 15% of all lung cancers. Radiotherapy (RT) plays a fundamental role in both early and advanced stages. Currently, in advanced disease, the use of consolidative chest RT should be recommended for patients with good response to platinum-based first-line chemotherapy, but its use has not yet been standardized. The present prospective study aims to evaluate the pattern of care of consolidative chest RT in patients with advanced stage SCLC, and its effectiveness in terms of disease control and tolerability. (2) Materials and methods: This study was a multicenter prospective observational trial, proposed and conducted within the AIRO lung study group to evaluate the pattern of care of consolidative chest RT after first-line chemotherapy in patients with advanced SCLC. The patient and tumor characteristics, doses, fractionation and volumes of thoracic RT and prophylactic cranial irradiation (PCI), as well as the thoracic and extrathoracic response to the treatment, toxicity and clinical outcomes, were collected and analyzed. (3) Results: From January 2017 to December 2019, sixty-four patients were enrolled. Median follow-up was 33 months. The median age was 68 years (range 42–81); 38 patients (59%) were male and 26 (41%) female. Carboplatin + etoposide for 6 cycles was the most commonly used first-line therapeutic scheme (42%). With regard to consolidative chest RT, 56% of patients (35) received 30 Gy in 10 factions and 16 patients (26%) received 45 Gy in 15 sessions. The modulated intensity technique was used in 84.5% of cases, and post-chemotherapy macroscopic residual disease was the target volume in 87.5% of patients. Forty-four patients (69%) also underwent PCI. At the last follow-up, over 60% of patients did not experience chest disease progression, while 67% showed extrathoracic progression. At the first radiological evaluation after RT, complete response and stable disease were recorded in 6% and 46% of the cases, respectively. Two patients had a long-term complete response to the combined treatment. The brain was the first site of extrathoracic progression in 28%. 1y and 2y OS and PFS were 67%, 19%, 28% and 6%, respectively. Consolidative chest RT was well-tolerated in the majority of patients; it was interrupted in three cases (due to G2 pulmonary toxicity, disease progression and clinical decay, respectively). Only 1 patient developed G3 asthenia. (4) Conclusions: Consolidative chest RT has been shown to be useful in reducing the risk of thoracic disease progression and is absolutely well-tolerated in patients with advanced stage SCLC with good response after first-line chemotherapy. Among the Italian centers that participated in this study, there is still variability in the choice of fractionation and target volumes, although the guidelines contain clear recommendations. The aim of future research should be to clarify the role and modalities of chest RT in the era of immunotherapy in advanced-stage SCLC.

## 1. Introduction

In the universe of lung cancer, approximately 15% is small cell lung cancer (SCLC) [1]. Seventy percent of patients already have advanced disease at diagnosis; this is due to the rapid growth and marked ability to develop early metastases of this disease [2]. This impacts a worse prognosis. Even today in clinical practice, an old classification of disease is still used that distinguishes SCLC into limited (LD-SCLC) and extensive (ES-SCLC) disease [3,4]. For many years, the combination etoposide + platinum (EP) was the standard first-line of treatment for extended forms at diagnosis, with an objective response rate of 40–70% [5]. Immune checkpoint inhibition (ICI) in the treatment of SCLC has been the focus of studies in recent years. The reason for pursuing the combination of new immunotherapy drugs with chemotherapy in SCLC is the high presence of tumor mutation, which can lead to a possible increased of immunogenicity. Chemotherapy may play a role in stimulating the expression of humoral antigens, making the tumor more responsive to checkpoint inhibitor therapy. Inhibition of CTLA-4 by immune checkpoint inhibitors and inhibition of PD-L1 have shown favorable results in terms of overall response rate (ORR), progression-free survival (PFS), and overall survival (OS) in this patient setting [6,7,8,9,10,11,12,13]. The combination of immunotherapy and first-line chemotherapy has shown positive results on OS and PFS as demonstrated by three phase III trials (IMpower133, CASPIAN and KEYNOTE-604), with a low toxicity profile [8,9,14,15,16].

To date, the discordance between patients enrolled in clinical trials and patients treated in real-life experience with immunotherapy is still evident; in fact, data on the safety and efficacy of anti-PD-1/anti-PD-L1 agents are still limited. All this results in a percentage of patients being excluded from these treatment regimens because of their performance status and because of side effects [17]. Phase III clinical trials with larger eligibility criteria and greater inclusion are probably needed to narrow this gap between reality and clinical trials. In terms of therapy, at all stages of disease presentation, radiotherapy (RT) plays a key role. In fact, for example, in the early stage, concurrent thoracic RT and platinum-based chemotherapy has become the standard of care, leading to increased OS [18,19]. In addition, prophylactic cranial irradiation (PCI) is suggested after thoracic treatment for locoregional stages, because the brain is often the site of disease progression [20]. More discussed is the role of RT in advanced disease. Regarding PCI, the Japanese phase III trial that randomized patients with stage IV SCLC to PCI (25 Gy of 2.5 Gy per day) showed that PCI reduced the appearance of brain metastases (32.9% vs. 59%), though did not show impact on OS (48% vs. 54% at 1 year) [21]. These results led to the drop-out of PCI in many cancer centers. In addition to cerebral disease management, intrathoracic tumor control after first-line treatment rests critical, as most patients with SCLC have persistence of thoracic disease, with disease progression within 1 year. The CREST study, a randomized phase III trial, evaluated the impact of thoracic RT after chemotherapy and PCI in advanced SCLC [22]. In this study, 498 patients with advanced SCLC randomized to exclusive PCI or PCI + thoracic RT (PCI + RT) after first-line chemotherapy demonstrated an increase in the 2-year survival rate of the PCI +RT group by 13%. Furthermore, the thoracic recurrence rate was 20% in the experimental arm and 46% in the standard arm. Thoracic RT was performed with a schedule of 30 Gy in 10 fractions. The authors’ conclusion was that chest RT should be recommended for patients with stage IV disease after first-line CHT chemotherapy. The mentioned conclusions are not yet applied in clinical practice, as to date there are not clear definitions regarding doses and volumes of irradiation in these patients.

This prospective study was carried out with the intention of evaluating the spread in Italy of the use of thoracic RT in ES-SCLC, with a focus on the pattern of care (RT modalities, volumes and doses) and its effectiveness in terms of disease control and tolerability.

## 2. Materials and Methods

This study was a multicenter prospective observational trial, proposed and conducted within the AIRO thoracic oncology study group, to evaluate the pattern of care of consolidative chest RT in patients with ES-SCLC respondent to first-line chemotherapy.

All participants had to meet the following criteria to be eligible for the study: (1) aged 18 years or older; (2) histological diagnosis of advanced small cell lung cancer (due to thoracic extension of disease, or to the presence of distant metastases) at onset; and (3) patients who underwent first-line platinum-based chemotherapy (for 4–6 cycles), obtaining good response (CT and/or CT-PET) (complete or partial response and stable disease, according to RECIST criteria [23]). Exclusion criteria were: (1) diagnosis of non-small cell lung tumor; (2) patients with limited SCLC disease; (3) patients unfit for first-line platinum-based treatment; (4) patients with progressive disease after first-line chemotherapy; (5) previous chest radiation therapy; and (6) patients diagnosed with other neoplasms in the previous 5 years (with the exception of non-melanomatous skin cancers).

### 2.1. Objectives

The primary objective was to describe the role of consolidative thoracic RT and the patterns of treatments delivered in eight Italian radiotherapy centers, in terms of total dose delivered, dose per fraction, treatment volumes and techniques in patients affected with ES-SCLC with partial response (PR), complete response (CR) or stable disease (SD) after first-line chemotherapy. The locoregional response rate was also evaluated.

Secondary endpoints were overall survival (OS), progression-free survival (PFS) and toxicity.

### 2.2. Data Collection

The patient and tumor characteristics, doses, fractionation and volumes of thoracic irradiation and PCI, as well as the thoracic and extrathoracic response to the treatment, treatment toxicity, according to CTCAE v4.0, and clinical outcomes were collected from each center in an online database, from which we extracted the results of the study. For the evaluation of toxicity and clinical response, a clinical visit and a restaging computed tomography (CT) scan were requested from each patient every three months.

### 2.3. Statistical Analysis

In absence of a priori hypothesis, given the exploratory nature of the study, no formal sample size calculation was performed; mainly for feasibility reasons, we analyzed data regarding 64 patients. Clinical and demographic data were expressed in terms of frequency and percentage for categorical variables, median and IQR for quantitative variables.

Overall survival (OS) was measured from diagnosis date to death (event) or last follow up; progression free survival (PFS) from diagnosis to death or any relapse, whichever comes first (event), or last follow up. Survival functions were estimated using the Kaplan–Meier method; their confidence intervals were two-tailed and calculated considering a 0.95 confidence level. Statistical analysis was performed using R 4.2.1 (R Core Team (2022). R. a Languageand environment for statistical computing. R Foundation for statistical Computing, Vienna, Austria).

### 2.4. Ethics Approval and Consent to Participate

The present study received final approval by our Institutional Ethics Committee “Comitato Etico Provinciale di Reggio Emilia, Azienda Ospedaliera IRCCS di Reggio Emilia”, protocol code: n° 154/2017, and was conducted in accordance with the principles of good clinical practice (GCP) in compliance with the ICH GCP guidelines and the ethical principles contained in the Declaration of Helsinki and its subsequent updates. A written consent form was obtained from each patient.

## 3. Results

From January 2017 to December 2019, sixty-four patients were enrolled. No screening failures were reported. Median follow-up was 33 months. Eight Italian centers were involved. The characteristics of the study population are summarized in Table 1.

The median age of the patients was 68 years (range 63–74); 38 (59%) patients were male and 26 (41%) female. Most presented with a performance status assessed according to the ECOG scale of 0–1; only five (8%) patients had a worse ECOG score of 2. Two patients had no smoking history, while 38 (59%) were active smokers, with a median of 40 packs/year. Carboplatin + etoposide for 6 cycles was the most commonly used first-line therapeutic scheme (42%). Seven patients discontinued chemotherapy due to adverse effects, mainly due to hematological toxicity (3 pts), allergic reaction (1 pt) or intolerance (1 pt). At the end of the chemotherapy treatment, most patients showed stable disease (SD) or partial response (PR) of both thoracic and extrathoracic disease, according to RECIST criteria [23]. Sixteen (10%) and three (5%) patients showed complete extra and intrathoracic response after chemotherapy, respectively.

As regards radiotherapy treatment, all 64 enrolled patients did receive consolidative thoracic RT. However, there was a large variability in terms of doses, treatment volumes and techniques. The most widely used fractionation was 3 Gy × 10 fr in 35 patients (55%), then 3 Gy × 15 fr in 16 (25%), while conventional fractionation (1.8–2 Gy/day) was used in only 5 patients (8%). The intensity modulated technique was used in 84.5% of cases, and post-chemotherapy residual disease was the target volume in 87.5% of patients. Three patients who had radiological CR (by RECIST criteria) after chemotherapy received consolidative RT to suspicious persistent subcentimeter mediastinal nodes, based on pre-chemotherapy imaging that was then co-registered with RT planning CT-scan. PCI was administered to 44 patients (69%), and the 2.5 Gy × 10 fractions scheme was the most commonly used. All patients received a brain staging with CT scan. A subsequent confirmation with MRI was required. No patient received metastasis directed ablative or palliative RT to metastatic lesions inside or outside the thorax.

After consolidative RT, over 66% of patients did not experience thoracic (“in-field”) progression of disease, while 67% experienced extrathoracic PD (Tab 2). In particular, the rates of CR, PR and SD were 6%, 14% and 46%, respectively. Two patients had a long-term CR after the combined treatment. The brain was the first site of extrathoracic progression (28%), followed by the adrenal gland, supraclavicular lymph nodes, liver and bone. At the time of writing, 23 patients had died due to distant progression of disease. One and two-year survival estimates were 67% and 19% (for OS), 28%, and 6% (for PFS), respectively. Median OS and PFS were 15.5 months (95%CI: 12.8–17.6) and 10.7 months (95%CI: 9.9–11.5), respectively. Figure 1 and Figure 2 show OS and PFS survival functions. The time to progression (TTP) was 10.7 months (95%CI: 10.1–11.5) in the whole study population. OS and DFS at 4 months was 100%.

Figure 1 and Figure 2 show the curves of OS and PFS.

Finally, considering toxicities, consolidative RT was well-tolerated in the majority of patients. Thoracic RT was interrupted only in three cases, due to G2 pulmonary toxicity, disease progression and clinical decay, respectively. Only one patient developed G3 asthenia, while no other cases of G3–4 acute toxicity were recorded. Table 2 summarizes the recorded RT-related acute toxicity.

## 4. Discussion

Small cell lung cancer is very responsive to chemotherapy, and quick response to such treatment is not unusual. Platinum-based chemotherapy, four to six cycles of cisplatin/carboplatin + etoposide or platinum + irinotecan is the standard first-line treatment in ES-SCLC. Carboplatin is generally preferred over cisplatin. However, most patients relapse in the following way: during CHT (platinum resistant), within 90 days (platinum refractory), or over 90 days (platinum sensitive) [24].

Intrathoracic tumor control after chemotherapy remains an issue in ES-SCLC, as most patients have SD, with progression within one year and negatively affecting the quality of life. Beneficial effects of chest RT were reported in a retrospective study by Zhu et al., who reported an advantage in terms of OS in 60 patients treated with a dose ranging from 40 to 60 Gy, at 1.8 to 2.0 Gy per fraction [25]. A recent retrospective study investigated the practice patterns for the radiation time and dose/fractionation of chest RT and attempted to identify prognostic factors for patients who would benefit from this approach [26]. The authors showed that patients treated with thoracic RT, in association with chemotherapy, experienced improved median OS (18.1 vs. 10.8 months), median PFS (9.3 vs. 6 months) and median local recurrence-free survival (12 vs. 6.6 months). In addition, it seems that early thoracic RT tended to prolong PFS, but not OS.

Moreover, in a randomized phase III study, after CHT plus PCI, chest RT (30 Gy in 10 fractions) was administered, reporting an OS at 2 years of 13%, versus 3% in the control group [22]. The authors concluded that chest RT should be considered for patients with any response to CHT. This treatment strategy was advocated for certain ES-SCLC patients, both in the 2020 NCCN guidelines and in the ASTRO 2020 guidelines. In routine practice this recommendation has not been applied. This may be due in part to the CREST trial demonstrating a 2-year OS benefit, but not a 1-year OS benefit, which was the primary endpoint of the study. Another possible reason for inconsistent use of consolidative thoracic RT is conflicting data, particularly a study by the Radiation Therapy Oncology Group, RTOG 0937, which did not demonstrate a survival benefit of consolidative chest RT combined with irradiation of distant metastatic sites [27]. Not only that, but the absence of clear guidelines determines a wide variability in terms of doses, volumes and treatments between the different RT centers. This data is also confirmed in the present study, created within the Italian AIRO thoracic oncology group, with the aim of studying the pattern of care in this patient setting. Our data shows a wide variability of RT doses. The RT protocol suggested by Slotman et al. [22] is the scheme most widely used also by Italian centers, and with the exception of a few patients, a hypofractionated regimen was preferred to conventional treatment (9%). Another issue in this patient setting is the definition of the irradiation fields, i.e., whether to include all pre-chemotherapy chest disease, as indicated in the treatment of LD-SCLC, or to limit oneself to irradiation of residual disease after chemotherapy treatment. In our case series, the second option was the preferred choice in most patients. In our opinion, a limited field approach is recommended to reduce the expected toxicity; however, in selected cases, such as young patients, patients with excellent performance status, limited disease volumes and excellent response to chemotherapy, an extended irradiation field can be considered. Our results confirm what emerged from the literature, namely, that the addition of RT makes it possible to consolidate the thoracic control of the disease obtained with chemotherapy for one year in most patients. In our series, RT maintained stable or reduced chest disease in 60% of cases and, in four patients, resulted in a complete response. Instead, most patients showed progression of extrathoracic disease, confirming the sensitivity of SCLC to RT. Not only does RT allow good control of the disease for a given time, but it also does not lead to severe side effects, with mild toxicity rates. The role of PCI is still an open question and a source of debate. The brain is often the organ of metastasis in patients with SCLC, and PCI is suggested in patients after first-line chemotherapy for a limited stage [20]. Patel published a meta-analysis that the incidence of brain metastasis decreased by more than 25 percent, at 3 years after PCI, with a doubling of survival: 42% versus 23%, at 2 years (*p* < 0.001) [28]. In a Japanese study, patients with stage-extended disease with any response to CHT were randomized to PCI or surveillance, with time to symptomatic metastasis as the primary endpoint [2]. Patients in the PCI arm had a lower risk of brain metastasis at 12 months. The 1-year survival rate was 27.1% in the irradiated group and 13.3% in the control group (*p* < 0.001). In this Japanese study, patients without negative brain MRI were enrolled and randomized to PCI (25 Gy in 10 daily fractions) or no other treatment. Subsequently, it was noted at the 1-year follow-up that PCI reduced the appearance of brain lesions (32.9% vs. 59%), however, it did not impact on OS (48% vs. 54% at 12 months),which is why, currently, international guidelines support PCI in patients who had a response after first-line CHT. The scheduled radiation dose is 25 Gy of 2.5 Gy per die, without sparing the hippocampus [22].

Considering the difficulty in clearing the question on PCI independently of the stage of disease, it has been suggested that a viable route is watchful waiting with periodic brain MRI, because it is certainly true that most patients do not have brain metastases at the time of diagnosis, but we also know how frequently the SCLC gives them, causing worse prognosis, increased morbidity and decreased quality of life. From the current data available to us, the efficacy of systemic chemotherapy for asymptomatic and symptomatic brain metastasis is unfortunately exiguous, as even maintenance chemotherapy has not been shown to reduce the incidence of later metastasis after first-line treatment. Perhaps the potential benefit of PCI should be read, evaluated and used as a weapon on the time of occurrence of brain metastases and, consequently, their impact on quality of life, rather than as OS.

In recent years, a real revolution has been happening in the field of lung pathology, in both non-small cell lung cancer and SCLC. In the field of RT, the use of stereotactic body radiation therapy (SBRT) [29] for the treatment of oligorecurrence/oligoprogression [30,31], rather than in reirradiation [32], has allowed greater local control of the disease and the postponement of second-line chemotherapy treatments, which are less effective than systemic upfront treatments.

The recent immunotherapy revolution that has affected the entire field of oncology has also manifested itself in ES-SCLC. Several studies [9,10,11,12,33,33] have shown important benefits in the use of immunotherapy. Among others, the IMPOWER-133 study reported improved OS (33.5% vs. 20.4% for placebo) with a median OS of 12.3 months in the atezolizumab group, and 10.3 months in the placebo group (hazard ratio for death, 0.70; 95% confidence interval [CI], 0.54 to 0.91; *p* = 0.007) [6,34], with a comparable safety profile to chemotherapy alone without impairment of quality of life [35]). The median PFS was 5.2 months and 4.3 months, respectively (hazard ratio for disease progression or death, 0.77; 95%CI, 0.62 to 0.96; *p* = 0.02).

Another phase III study (the CASPIAN trial) evaluated the effectiveness of the addition of durvalumab and tremelimumab to CHT with metastasized small cell lung cancer [9]. Patients were randomly assigned (in a 1:1:1 ratio) to durvalumab plus chemotherapy, durvalumab/tremelimumab plus chemotherapy, or platinum/etoposide alone. Durvalumab plus CHT significantly improved OS, with a hazard ratio of 0.73 (95% CI 0.59–0.91; *p* = 0.0047); median OS was 13.0 months (95%CI 11.5–14.8) in the durvalumab plus platinum–etoposide group versus 10.3 months (9.3–11.2) in the platinum–etoposide group, with 34% (26.9–41.0), versus 25% (18.4–31.6) of patients alive at 18 months. Median PFS was 5.1 months (95%CI 4.7–6.2), with durvalumab plus platinum–etoposide, versus 5.4 months (4.8–6.2) with platinum–etoposide; the 6-month PFS rates were 45% (39.3–51.3), versus 46% (39.3–51.7); and the 12-month PFS rates were 18% (13.1–22.5), versus 5% (2.4–8.0). No additional benefit of tremelimumab was observed [9]. In contrast, excellent long-term results were seen in patients treated with durvalumab + CHT, versus CHT alone; three times as many patients achieved long-term benefits with the therapeutic combination [32]. Patients in all arms with PFS equal to 1 year had a better overall response rate (ORR), duration of response and OS than the subgroup with PFS < 1 y. Furthermore, in the phase III KEYNOTE-604 study, OS was longer in the pembrolizumab + chemotherapy arm, compared with chemotherapy alone [15]. Although PFS improved in the pembrolizumab arm, the significance threshold was not reached (HR 0.8, 95%CI 0.61–0.98). The ORR was 71% in the pembrolizumab arm, and 62% for placebo. In these cited studies, it was definitively noted that the coupling of immunotherapy to chemotherapy did not reduce quality of life. In contrast, however, in the randomly assigned phase III IDEATE trial, where the patient could receive chemotherapy (platinum/ethoposide) + ipilimumab or placebo, the primary endpoint OS was not achieved, revealing a higher rate of adverse events in the CHT plus ipilimumab arm [11]. In consideration of the results achieved by the aforementioned three randomized phase III trials, where the addition of immunotherapy to first-line chemotherapy revealed improved OS and PFS with a safety profile and QoL, this treatment was allowed to be considered as a current standard of care. However, none of these studies included consolidative chest RT in the enrolled patients. Also, less comforting were the results of the IMpower133 and CASPIAN studies, where the assessment of the therapeutic effect of immunotherapy was modest in terms of median OS, improving by 2–3 months in the experimental arms of both studies.

If we compare the median survivals of these studies with those reported in our study, we note that, in our experience, they are better in terms of both median OS and median PFS, as shown in Table 3.

Despite the great limitations of this comparison, the question is still valid whether the use of consolidative RT can contribute to improving the prognosis of these patients, even, and especially, in the era of immunotherapy. In our opinion, it is plausible that some ES SCLC patients could derive additional benefits from thoracic RT in association with chemo-immunotherapy, and that future trials should go in this direction. Furthermore, there is a very strong biological rationale for combining immunotherapy with CHT in SCLC, which is the high mutations, with enhanced immunogenicity. CHT may stimulate the expression of tumor antigens, priming the tumor for response to checkpoint inhibitory therapy. Moreover, pre-clinical data show evidence of a synergistic immune stimulation against cancer cells in incorporating radiotherapy and immunotherapy [38]. There are currently no studies evaluating the use of consolidative RT after first-line chemo-immunotherapy; however, given the premises based on preclinical evidence and the advantage of RT in local control of SCLC, its use in this emerging setting is recommended in clinical practice.

From an analysis of the literature, one of the most important retrospective experiences of consolidative chest RT after first-line chemo-immunotherapy for ES-SCLC was reported by Gross et al. [39]. A total of 244 patients with ES-SCLC were analyzed in this study, of which 63 received consolidative chest RT and 181 patients did not. After propensity score matching by age, gender, race, comorbidity score and type of chemotherapy, there was a trend towards improved median survival from 9 to 11 months and 2-year OS at 18.1%, with the addition of chest RT vs. 12.0% with chemo-immunotherapy alone (*p* = 0.067). The authors concluded that in patients with ES-SCLC, the addition of thoracic RT to chemo-immunotherapy is associated with a trend towards improved median OS. The increase in 2-year OS shown in this study appears to be of the same order of magnitude reported by the randomized studies since the addition of immunotherapy to chemotherapy, despite the non-randomized nature of the study and the limitation of the small sample size.

Important data have recently been published on the safety of the combination of immunotherapy and chest irradiation in ES-SCLC. MD Anderson conducted a phase I study of 38 patients that demonstrated that thoracic RT and concomitant pembrolizumab were well-tolerated, with no grade 4/5 toxicity, and only 6% experienced grade 3 toxicity [40]. In addition, the use of nivolumab plus ipilimumab in combination with chest RT (30 Gy in 10 fractions) after standard platinum-based chemotherapy was evaluated in a phase I/II study, initially with 52 patients with ES-SCLC enrolled. This study was to be conducted in two phases. The aim of the first phase was to establish the recommended dose of immunotherapy in combination with chest RT for the phase II study, while the aim of the second phase was an evaluation on PFS. During a planned interim analysis, the authors reported immune-related adverse events (IRAEs) possibly or definitely attributable to study therapy, with a 19.1 percent rate of grade 3 or higher pulmonary IRAEs. The estimated PFS at six months was 24%, thus not significantly improved over historical controls, which is why, after enrollment of only 21 patients, the trial was stopped. However, the toxicity profile of immunotherapy and thoracic RT was in line with the toxicity normally attributed to nivolumab and ipilimumab, which suggests that thoracic RT does not increase the risk of serious toxicities [41]. Recently, a multi-institutional retrospective analysis was published with the aim of evaluating outcomes and toxicities after first-line atezolizumab plus chemotherapy followed by chest RT in patients with ES-SCLC [42]. This retrospective analysis was conducted by enrolling 20 patients, with a median follow-up of 12 months; the median OS was 16 months with a 1-year OS of 77.5% (comparable to modern clinical trials including CREST, IMpower133 and CASPIAN). This study showed a very favorable safety profile with low toxicity rates (0% grade 3+ toxicity, 0% grade 2 pneumonia and 5% grade 2 esophagitis) with this approach. These results regarding toxicity rates are comparable to the results of the CREST study, which reported 1.2% grade 3 dyspnea and 1.6% grade 3 esophagitis with chest RT after first-line chemotherapy. A phase II/III trial (NRG Oncology LU007) is still ongoing to evaluate consolidated thoracic RT to the chest and extra-thoracic sites after 4–6 cycles of carboplatin/etoposide/atezolizumab in patients with ES-SCLC. In this study, patients will be randomized to RT (up to a maximum of 5 sites) + atezolizumab vs. maintenance atezolizumab alone. In this study, the authors provide the option of PCI according to their discretion. The study just described will be based on data from 300 patients—data that will be very useful to understand the actual role that thoracic RT and more can play in the era of chemo-immunotherapy. It will be imperative to continue to collect data derived from prospective, randomized trials to understand precisely which patients will benefit from thoracic RT in the era of immunotherapy, but, to date, we believe that consolidative thoracic RT should be offered to patients who show a good response after chemo-immunotherapy, particularly in whom with residual thoracic disease after first-line therapy.

The advantage of PCI in association with immunotherapy is very controversial. To our knowledge, there are no studies that support a clear benefit of adding PCI with immunotherapy. In the Impower-133 study [34], only 11% (22 patients) received PCI, while the CASPIAN trial [9] only allowed PCI on the control arm, of which 8% (21 patients) received it. Subgroup analyses of these trials suggest some intracranial efficacy of immunotherapy. No firm recommendations can be made on the use of PCI in ES-SCLC, given the conflicting evidence. The benefits and toxicities of PCI need to be weighed up against the resources and costs required for MRI surveillance, and patient age and risk factors for cognitive decline are also important [43].

The present study, despite its prospective observational nature, is limited by the small sample size, reflecting the fact that thoracic RT after first-line chemotherapy has not been uniformly adopted across the RT centers in patients with ES-SCLC. The results of our study confirm the great variability of total doses, fractionation and treatment volumes, as well as effectiveness in reducing the risk of intrathoracic disease progression. It is obvious that these results are not applicable in the current reality where the first-line therapy is chemo-immunotherapy, but it further underlines the need for standardization of first-line therapy in patients with ES-SCLC.

## 5. Conclusions

Consolidative chest RT has been shown to be useful in reducing the risk of thoracic disease progression and is absolutely well-tolerated in patients with ES-SCLC with good response after first-line chemotherapy, even if, among the Italian centers that participated in this study, there is still variability in the choice of fractionation and target volumes, although the guidelines contain clear recommendations. Though chemotherapy remains the mainstay of ES-SCLC treatment and the combination with immunotherapy has led a significant improvement in outcomes, we believe that thoracic RT may still play an important role in the management of ES-SCLC patients. Indeed, given that chest RT is well-tolerated and has previously demonstrated a survival advantage after chemotherapy, it may be reasonable to consider whether adding it to chemo-immunotherapy may lead to an even more substantial benefit. A number of current clinical trials are investigating immune-radiation therapy, also in the ES-SCLC setting, but safety data on this association is already available in both the SCLC and NSCLC settings. While waiting for the results of these studies, we believe that chest RT should be offered to patients with residual chest disease after chemo-immunotherapy.

## Figures and Tables

**Figure 1 cancers-15-00434-f001:**
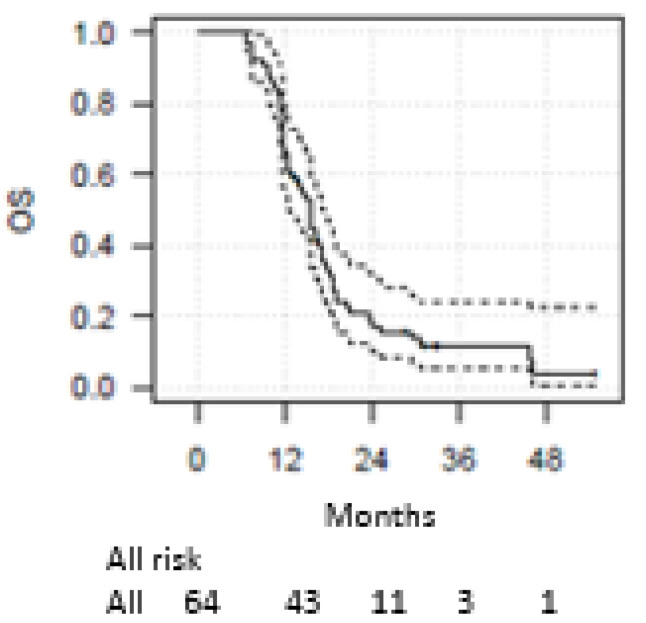
Overall Survival.

**Figure 2 cancers-15-00434-f002:**
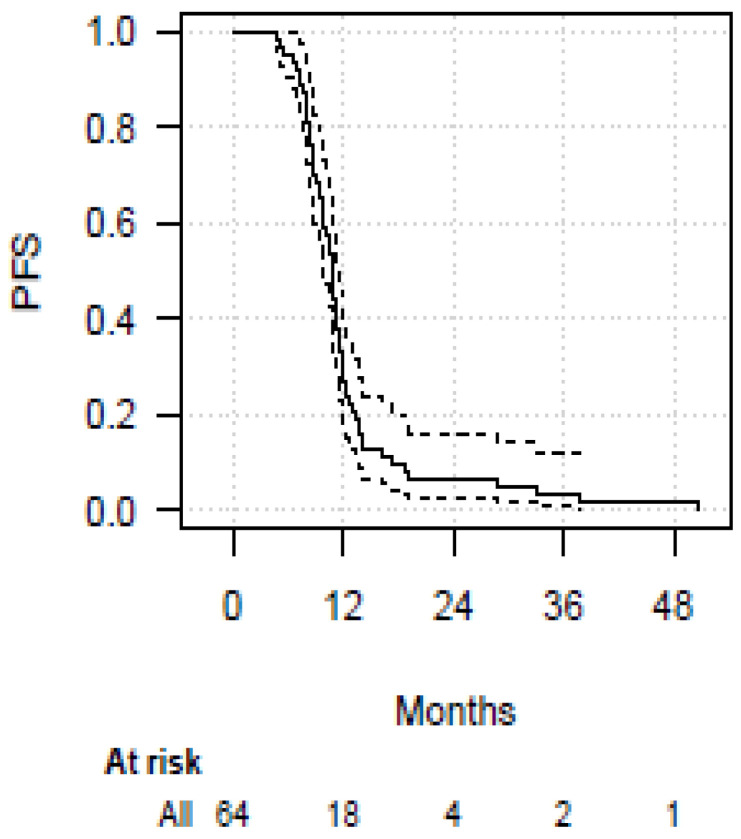
Progression-Free Survival. Overall survival and progression-free survival Kaplan–Meier curves (solid line), with 95% confidence interval (dashed line).

**Table 1 cancers-15-00434-t001:** Characteristics of the study population.

Parameter		N (%)
Age (years)		
	Median	68 (63–74)
	Range	42–81
Sex		
	Female	26 (41)
	Male	38 (59)
BMI		
	Median	25 (22.7–26.2)
ECOG		
	0	34 (53)
	1	25 (39)
	2	5 (8)
Chemotherapy scheme		
	carboplatin + etoposide	42 (65.5)
	cisplatin + etoposide	22 (34.5)
Dose per fractions		
	3 Gy × 10 fr	35 (55)
	1.8 Gy × 33 fr	3 (5)
	3 Gy × 15 fr	16 (25)
	2 Gy × 30 fr	2 (3)
	4 Gy × 5 fr	3 (5)
	3 Gy × 13 fr	5 (7)
RT technique		
	3DCRT	10 (15.5)
	IMRT	54 (84.5)
RT Volume		
	pre-cht	8 (12.5)
	post-cht	56 (87.5)
PCI		
	Yes	44 (69)
	No	20 (31)
PCI dose		
	4 Gy × 5 fr	7 (16)
	2.5 Gy ×10 fr	29 (66)
	3 Gy × 10 fr	8 (18)

Abbreviations: BMI: Body mass index; ECOG: Eastern Cooperative Oncology Group; Gy: gray; fr: fractions; RT: radiotherapy; CT: chemotherapy; PCI: prophylactic cranial irradiation; IMRT: intensity modulated radiotherapy; 3DCRT: 3-dimensional conformal radiotherapy.

**Table 2 cancers-15-00434-t002:** Acute radiotherapy-related toxicity, according to CTCAE v4.0.

Toxicity	Grade	N (%)
Pulmonary		
	G0	54 (84)
	G1	6 (10)
	G2	4 (6)
	>G3	0
Esophagus		
	G0	39 (61)
	G1	20 (31)
	G2	5 (8)
	>G3	0
Asthenia		
	G0	61 (95.5)
	G1	1 (1.5)
	G2	1 (1.5)
	G3	1 (1.5)
	>G4	0

Abbreviation: N: number; G: grade.

**Table 3 cancers-15-00434-t003:** Comparison between major I-line immuno-chemotherapy trials and thoracic consolidation studies in ED-SCLC.

Study [Ref.] (Experimental Arm)	TRIAL DESIGN	Median OS (Months)	Median PFS (Months)
IMpower133 [34]	Phase 3, RCT, double-blind	12.3	5.2
CASPIAN [9]	Phase 3, RCT, open-label	13	5.1
KEYNOTE-604 [10]	Phase 3, RCT, double-blind	10.8	4.5
IDEATE [11]	Phase 3, RCT, double-blind	11.0	4.6
NCT01331525 [33]	Phase 2, RCT, double-blind	17.0	6.9
ECOG-ACRIN EA5161 [36]	Phase 2, RCT, double-blind	11.3	5.5
REACTION [37]	Phase 2, RCT	12.3	5.4
CREST [22]	Phase 3, RCT, open-label	8	4
TRENDS [our study]	Prospective single arm	15.5	10.7

## Data Availability

The data of this work can be disclosed by mail contact with the correspondent and with the legal office.

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
