# Peer review of "Thoracic Radiotherapy in Extensive Disease Small Cell Lung Cancer: Multicenter Prospective Observational TRENDS Study"

_cancers, 2023, doi:10.3390/cancers15020434_

Round 1
Reviewer 1 Report
The authors here describe the use of consolidative radiation therapy in ES SCLC in 8 centers in Italy. The rationale for the study is that consolidative thoracic radiation therapy is somewhat controversial and has not been widely adopted since the original publication demonstrating a survival benefit. Thus, the authors aimed to describe the use of thoracic radiotherapy in Italy in a prospective observational study.
It is a little difficult to understand the purpose of this study. It is not hypothesis-testing really as there is no hypothesis. It is merely a description of how the use of consolidative radiotherapy has been adopted.
From a study design perspective, therefore, it would have been nice to dive deeper into the practices of physicians - try to understand why certain choices were made (fractionation schedule, dose, why was radiation chosen or not chosen).
It would be useful to have a consort diagram to understand how patients were screened, included, excluded. The 64 patients - is this the number who were included because that was the number of eligible patients, or were there other reasons patients were excluded.
What happened to patients who fit the eligibility criteria but did not get radiotherapy? (As per methods eligible patients were not required to recieve radiotherapy - but all patients included did get radiotherapy?)
In patients who experienced a CR from chemotherapy, was radiation therapy used? If so, why and to where?
If patients had residual extrathoracic disease, did they receive consolidative thoracic radiotherapy?
This data herein add little new information to help us decide the utility of this approach. The authors describe that there is wide variability in the use of thoracic radiotherapy amongst the different sites. There is only 64 patients which is a rather small sample size given that the heterogeneity of the approach. The results are not unexpected in any way - that is, the use of thoracic radiotherapy results in response in the chest but extrathoracic mestastasis is the main site of progression.
The authors correctly point out that the therapeutic landscape has now changed with the incorporation of immunotherapy and neither these data nor the prior published literature take into account the influence of immunotherapy on outcomes with radiotherapy.
Author Response
The authors here describe the use of consolidative radiation therapy in ES SCLC in 8 centers in Italy. The rationale for the study is that consolidative thoracic radiation therapy is somewhat controversial and has not been widely adopted since the original publication demonstrating a survival benefit. Thus, the authors aimed to describe the use of thoracic radiotherapy in Italy in a prospective observational study.
It is a little difficult to understand the purpose of this study. It is not hypothesis-testing really as there is no hypothesis. It is merely a description of how the use of consolidative radiotherapy has been adopted.
Dear Reviewer, thank you very much for your comment.
We confirm that the study was not “hypothesis generating” because the efficacy of consolidative RT was already demonstrated in the CREST study. However, the use of thoracic RT has not been widely adopted due to different reasons: logistic concern, not clear evidence of overall survival benefitin following studies, lack of information on dose and volume to well balance benefit and side effects, no communication between medical and radiation oncologist and so on. So the aim of our study is not to describe the use of RT, but to better understand the role of consolidative RT in patients prospectively enrolled using a very pragmatic method such as single arm observational trial. In this way all patients eligible due to inclusion criteria were evaluated and then they could have been submitted to RT obtaining a homogeneus population that may help us to obtain more info on clinical outcomes and safety. Unfortunately, randomized clnical trials (that may achieve more robust results) are very difficult to carry on in this setting due to the very select population, the rarity of the disease and the recent of introduction of IO. Furthermore, local progression still represents a very critical issue in SCLC because it often causes an evident deteration of patients’ quality of life due to symptoms appearance and very low responsiveness to second line approaches. For these reasons we think that the work may help clinicians in their daily work confirming the good results prevously obtained in clinical trials both in terms of clinical outcomes and safety.
From a study design perspective, therefore, it would have been nice to dive deeper into the practices of physicians - try to understand why certain choices were made (fractionation schedule, dose, why was radiation chosen or not chosen).
Thank you for your comment.
This is a single arm prospective observational clinical study to understand the role of consolidative RT in a particular setting such as ES-SCLC. Due to the lack of strong and clear information about the best dose, fractionation and volumes even in the main international literature we thought it could have been better to let participants free about that issue (following single Insititute protocol). On the other hand, in our opinion, to go deeper on the reasons why a Center choose a dose/schedule/volume instead of another is more likely typical of a survey or editorial or something similar, buti t was not the aim of our project.
It would be useful to have a consort diagram to understand how patients were screened, included, excluded. The 64 patients - is this the number who were included because that was the number of eligible patients, or were there other reasons patients were excluded.
I confirm that 64 was the total number of eligible patients and all of them received consolidative thoracic RT. After further check of the database and contacting each PI of the participating Centers we can confirm that no patients were excluded after being enrolled (no screening failure reported). We added this information in the “Results” section.
What happened to patients who fit the eligibility criteria but did not get radiotherapy? (As per methods eligible patients were not required to recieve radiotherapy - but all patients included did get radiotherapy?)
I confirm that 64 was the total number of eligible patients and all of them received consolidative thoracic RT. We added this information in the “Results” section.
In patients who experienced a CR from chemotherapy, was radiation therapy used? If so, why and to where?
Three patients who had complete response after chemotherapy received RT to the involved mediastinal nodes based on pre-chemotherapy imaging available that was then co-registrated with RT planning CT-scan. In all three patients, sub-centimeter suspicious mediastinal nodes were detected at the RT planning CT-scan representing our CTV. We added this information in the “Results” section.
If patients had residual extrathoracic disease, did they receive consolidative thoracic radiotherapy?
No patient received metastasis directed therapy on extrathoracic lesion concomitantly or consecutively to consolidative thoracic RT. The treatment of extrathoracic lesions was allowed only in case of appearance of symptoms (palliative RT) or further (oligo)progression. In case of sistemic polystational progression, sistemic second line treatment was the preferred choice.
This data herein add little new information to help us decide the utility of this approach. The authors describe that there is wide variability in the use of thoracic radiotherapy amongst the different sites. There is only 64 patients which is a rather small sample size given that the heterogeneity of the approach. The results are not unexpected in any way - that is, the use of thoracic radiotherapy results in response in the chest but extrathoracic mestastasis is the main site of progression.
Thank you for your comment.
Surely, a total of 64 patients don’ t represent a large cohort in general terms, but it is one of the largest one between the prospective studies actually available. Indeed, after the CREST publication and the following metanalysis most of published studies are retrospective and frequently monocentric. The streghtness of our project is represented by its prospective nature, rigorous inclusion criteria and follow up and finally the multicentric base (preferentially driven ikkkuby tertiary cancer center). Furthermore, the demonstration that consolidative thoracic RT may significantly lower the rate of local progression should be an interesting confirmation in the immunotherapy era. Indeed, confirming the role of thoracic RT may be useful to future approaches or clinical trials that may concern the combination of local and immuno-mediated treatments.
The authors correctly point out that the therapeutic landscape has now changed with the incorporation of immunotherapy and neither these data nor the prior published literature take into account the influence of immunotherapy on outcomes with radiotherapy.
Thank You.
Reviewer 2 Report
This is a well written paper to emphasize the importance of consolidation of Thoracic Radiation Therapy (TRT) with more modern techniques which reduced progression in the thorax with moderate l toxicities by the consolidation of the TRT. My recommendations are as follows.1)Introduction can be shorten. 2)Fig.1 (OS and PFS)requires the explanations of three lines in the legend.3) Reference 46 is missing from the page 14.
Author Response
This is a well written paper to emphasize the importance of consolidation of Thoracic Radiation Therapy (TRT) with more modern techniques which reduced progression in the thorax with moderate l toxicities by the consolidation of the TRT.
We are glad you enjoyed the article and thank you for your comments
My recommendations are as follows.
1)Introduction can be shorten.
It has been done. Thanks
2) Fig.1 (OS and PFS)requires the explanations of three lines in the legend.
We add “Overall survival an Progression Free survival Kaplan Meier curves (solid line) with 95% confidence interval (dashed line)”.
3) Reference 46 is missing from the page 14.
WE are very sorry for the mistake. We added reference 46. Thank you very much
Reviewer 3 Report
This is an interesting study on an important subject. Although it is a very small study, it provides insight in real life data on the use of thoracic radiotherapy in ES-SCLC. It is surprising to side the heterogeneity of schedules and techniques and gives a good overview of the literature.
Table 2 is confusing and headings “Response after consolidative radiotherapy” does not make much sense. It would be better to describe in detail when this was examined (eg. 6 weeks after last radiotherapy) and how many of the patients with an initial PR/SD/PD improved.
For survival comparisons as in Table 4, the authors should add that for instance the CREST study measured OS from time of randomization after chemo, so 4 months should be added for comparison with other studies.
Author Response
This is an interesting study on an important subject. Although it is a very small study, it provides insight in real life data on the use of thoracic radiotherapy in ES-SCLC. It is surprising to side the heterogeneity of schedules and techniques and gives a good overview of the literature.
We are glad you enjoyed the article and thank you for your comments
Table 2 is confusing and headings “Response after consolidative radiotherapy” does not make much sense. It would be better to describe in detail when this was examined (eg. 6 weeks after last radiotherapy) and how many of the patients with an initial PR/SD/PD improved.
Thank you. We deleted tab 2. we have described the results in more detail
For survival comparisons as in Table 4, the authors should add that for instance the CREST study measured OS from time of randomization after chemo, so 4 months should be added for comparison with other studies.
Dear review. You are right, however, there is a limitation to the actual comparison between studies, as in the CREST study the analysis of OS and DFS was from the time of randomization, whereas in our case it is from the end of RT. However, In consideration of this, we prefer to report the median data in the table, while we have added the 4-month data in the text. Thank you
Round 2
Reviewer 1 Report
The manuscript in its current form has been improved. I have no qualms about the way the study was conducted and the data are what they are. My fundamental critique of the manuscript has not changed, however. This dataset adds very little new information. I suppose the main conclusion is that thoracic radiotherapy in this real world data sort of confirm what was observed in the CREST trial. The main strength of this research is that it is prospective and in some ways real world since no restrictions were placed on the manner of completing radiotherapy.
I appreciate the clarification on the points listed above.
In that regard, I think it is acceptable in the current form.